# A Suboptimal Optimizing Strategy for Velocity Vector Estimation in Single-Observer Passive Localization

**DOI:** 10.3390/s23135940

**Published:** 2023-06-26

**Authors:** Shuyi Gu, Zhenghua Luo, Yingjun Chu, Yanghui Xu, Junxiong Guo

**Affiliations:** 1School of Electronic Information and Electrical Engineering, Chengdu University, Chengdu 610106, China; gushuyi@stu.cdu.edu.cn (S.G.); guojunxiong@cdu.edu.cn (J.G.); 2The Fifth Institute of Telecommunication Science and Technology, Chengdu 610036, China; chuyj@5ritt.com; 3Sichuan Time Frequency Synchronization System and Application Engineering Technology Research Center, Chengdu 610062, China; 4Chengdu Jinjiang Electronic System Engineering Company, Ltd., Chengdu 610051, China; jecdz@jec784.com

**Keywords:** single-observer passive localization, time of arrival (TOA), direction of arrival (DOA), optimization, Kalman filtering (KF), simulated annealing (SA)

## Abstract

In a single-observer passive localization system, the velocity and position of the target are estimated simultaneously. However, this can lead to correlated errors and distortion of the estimated value, making independent estimation of the speed and position necessary. In this study, we introduce a novel optimization strategy, suboptimal estimation, for independently estimating the velocity vector in single-observer passive localization. The suboptimal estimation strategy converts the estimation of the velocity vector into a search for the global optimal solution by dynamically weighting multiple optimization criteria from the starting point in the solution space. Simulation verification is conducted using uniform motion and constant acceleration models. The results demonstrate that the proposed method converges faster with higher accuracy and strong robustness.

## 1. Introduction

Single-observer passive localization [1] is a popular technology with low cost, high concealment, and mobility [2]. The velocity and position are estimated together in passive localization [3]; however, this causes their errors to affect each other. Estimating them separately reduces errors that significantly affect the accuracy of passive localization. Position and velocity have long been considered estimation problems in passive localization [4,5,6]. Due to the nonlinearity of the observation model, nonlinear filtering algorithms improved by a Kalman filter (KF) [7] are still the most economical and effective methods in passive localization [8]. The most widely used filter is the extended Kalman filter (EKF) [9], which linearizes the model before applying the KF. The unscented Kalman filter (UKF) [10,11] is more suitable for nonlinear models than the EKF because it reduces the error of linearization through the unscented transform (UT) [12]. Although these methods have worked well for multiple-observer passive localization [13,14], problems including linearization errors [15], ill-conditioned covariance matrices [16], and insufficient observation information can make them inaccurate for single-observer passive localization.

To address the shortcomings of the KF in nonlinear filtering problems, optimization methods have been integrated into the KF framework, including the particle swarm optimization (PSO) [17], genetic algorithm (GA) [18], arithmetic optimization algorithm (AOA) [19], and cuckoo search algorithm (CSA) [20]. These are optimization algorithms assisted by a Kalman filter. In the traditional KF framework, these methods consider the KF process as an optimization problem; the optimization process is used to solve the parameters to be estimated. Although these methods outperform traditional KF, they come at the cost of more complex calculations and algorithm implementation overhead.

In this study, we could only obtain time-of-arrival (TOA) [21] and direction-of-arrival (DOA) [22] observations in a single-observer condition, known as hybrid TOA-DOA [23,24]. From this model, we propose an optimization strategy based on the KF framework to overcome the issues faced with less observation information. We focused on the problem of velocity estimation in passive localization, which cannot be solved using these methods in a single-observer condition. Our method (suboptimal estimation) searches for the true value of the velocity vector by dynamically weighting two optimization criteria from the initial position in the solution space. To ensure convergence, we incorporated a simulated annealing (SA) [25] mechanism into the algorithm. We simulated two possible scenarios of single-observer passive localization in a uniform motion model with constant acceleration and compared the proposed method with traditional methods to demonstrate its effectiveness and superiority. The results show that our approach converges more quickly and accurately to the true value of the velocity vector, with strong robustness in acceleration disturbances.

## 2. Preliminaries

In the following section, we represent vectors using bold font (*X*) and scalars using regular font (*x*); ∥·∥ represents the two norms. The estimator is labeled with a hat symbol ($\hat{x}$); the true parameter is not.

### 2.1. Model and Observations

The single-observer passive localization scenario is shown in Figure 1, considering air and ground nodes; $v_{A}$ represents the speed of the air node, and the speed of the ground node is $v_{G}$. The azimuth angle of the two nodes is β, and the pitch angle is ε. When both nodes are moving, ideally, the ground node can obtain its relative velocity. According to the kinematic positioning principle, the relative positions of the two nodes can be calculated. In this study, we focused mainly on velocity estimation.

In Figure 1, the positions of the air and ground nodes can be expressed as
(1)$$\begin{array}{cc} \; & P_{k}^{A}={\left[\begin{array}{ccc} X_{k}^{A} & Y_{k}^{A} & Z_{k}^{A} \end{array}\right]}^{T}\\ \; & P_{k}^{G}={\left[\begin{array}{ccc} X_{k}^{G} & Y_{k}^{G} & Z_{k}^{G} \end{array}\right]}^{T} \end{array}$$

The relative velocity vector at *k* moment between two nodes can be expressed as
(2)$$v_{k}={\left[\begin{array}{ccc} v_{Ak}^{x}-v_{Gk}^{x} & v_{Ak}^{y}-v_{Gk}^{y} & v_{Ak}^{z}-v_{Gk}^{z} \end{array}\right]}^{T}={\left[\begin{array}{ccc} v_{k}^{x} & v_{k}^{y} & v_{k}^{z} \end{array}\right]}^{T}$$

The relative position between the two nodes at *k* moment can be represented by a radial vector, expressed as
(3)$$R_{k}=P_{k}^{A}-P_{k}^{G}={\left[\begin{array}{ccc} X_{k} & Y_{k} & Z_{k} \end{array}\right]}^{T}$$

In passive localization, the relative position cannot be directly observed. However, for passive nodes, changes in relative position can be observed. A common approach is to observe the periodic pulses emitted by a located node; the change in the pulse arrival time can reflect the variables of the radial vector [26]. When a passive node observes a change in the radial vector at constant intervals Δt, the change in the radial vector between the two nodes can be expressed as
(4)$$\begin{array}{c} ∥R_{k}-R_{k-1}∥=r_{k}=\cos\beta_{k}\cos\varepsilon_{k}v_{k}^{x}\Delta t+\sin\beta_{k}\cos\varepsilon_{k}v_{k}^{y}\Delta t+\sin\varepsilon_{k}v_{k}^{z}\Delta t\\ =\Delta tF_{k}v_{k} \end{array}$$
where $v_{k}$ denotes the velocity vector between two nodes at moment *k*. β and ε are the azimuth angle and the pitch angle of the two nodes, which can be determined by the amplitude or phase of the radiated pulse. *F* denotes the state transfer matrix. Its function is to project the components of the velocity vector on each basis in the radial vector direction; it is multiplied by the observation time interval to obtain the relative displacement of the two nodes in Δt.

Then, the state transfer equation of when Δt is considered for normalization can be established as
(5)$${\left.r_{k}\right|}_{\Delta t=1}\equiv F_{k}v_{k}$$

For $r_{k}$, β and ε, generally, only observed values can be obtained in the following form:(6)$$\left\{\begin{array}{c} {\hat{r}}_{k}=r_{k}+e_{r}\\ {\hat{\beta }}_{k}=\beta_{k}+e_{\beta }\\ {\hat{\varepsilon }}_{k}=\varepsilon_{k}+e_{\varepsilon }\\ {\hat{F}}_{k}=\left[\cos{\hat{\beta }}_{k}\cos{\hat{\varepsilon }}_{k}\sin{\hat{\beta }}_{k}\cos{\hat{\varepsilon }}_{k}\sin{\hat{\varepsilon }}_{k}\right] \end{array}\right.$$
where *e* is the observation error, which is generally a random variable that follows a Gaussian distribution. The measurement errors $e_{r}$, $e_{\varepsilon }$ and $e_{\beta }$ are considered random variables with zero mean and variance of ${\sigma_{r}}^{2}$, ${\sigma_{\beta }}^{2}$ and ${\sigma_{\varepsilon }}^{2}$.

### 2.2. Kalman Filtering

The Kalman filter can be used to obtain the minimum mean square error (MMSE) of the estimated covariance matrix of the velocity vector for the linearized model. In an iterative algorithm, the velocity vector of each iteration is the result of the previous iteration of the Kalman filter. Considering the relatively uniform motion between nodes and using Equation (Equation 4) as the observation model, we obtain
(7)$${\hat{v}}_{k}={\hat{v}}_{k-1}$$

The covariance matrix of velocity vector estimation error *P* is updated as
(8)$$P_{v_{k}}=P_{v_{k-1}}+Q$$
where *Q* is the covariance matrix of the process error. In the model considered in this study, where we assume that the two nodes are relatively uniform in speed, the process noise consists mainly of acceleration disturbances in each velocity component. The Kalman gain is calculated. *R* is the covariance matrix of the measurement error:(9)$$K=P_{v_{k}}{\hat{F}}_{k}^{T}{\left({\hat{F}}_{k}P{\hat{F}}_{k}^{T}+R\right)}^{-1}$$

The velocity vector can be corrected based on the Kalman gain *K* and observed values *r* using Equation (Equation 10):(10)$${\hat{v}}_{k+1}={\hat{v}}_{k}+K\left(r_{k}-{\hat{F}}_{k}{\hat{v}}_{k}\right)={\hat{v}}_{k}+K\left(r_{k}-{\hat{r}}_{k}\right)$$

In Equation (Equation 10), the error between the predicted value ${\hat{r}}_{k}$ and observation value $r_{k}$ is known as the prediction error. When a velocity–vector estimation value close to the true value is used, the prediction error becomes very small. However, in this model, there is a condition in which the estimated velocity vector is far from the true value, but the prediction error is still very small; thus, the velocity vector cannot be accurately estimated. This study aims to improve this condition.

The covariance matrix of error *P* must be corrected simultaneously because the error distribution changes with the correction of the velocity vector:(11)$$P_{v_{k+1}}=\left(I+KF_{k}\right)P_{v_{k}}$$

Using the Kalman filter to estimate the velocity vector requires repeating the iterative process in Equations (Equation 7)–(Equation 11). When the prediction error remains almost unchanged, the Kalman filter is converged.

Although the KF is an optimal estimation method when the various errors mentioned above follow a Gaussian distribution, there may be divergence or distortion due to errors or initial values [27] in the linearization of the nonlinear models. In addition, in this study, the number of observations is limited; we believe that the prior knowledge of the velocity vectors is not necessarily accurate. To address these issues, improved algorithms such as the EKF, UKF, and MVEKF have been proposed. However, when applied to the proposed observation model, these methods do not significantly address these problems. Figure 2 illustrates the error of the velocity vector estimation using the EKF and UKF when the initial value deviates from the true value; $\delta_{v_{0}}$ represents the difference between the initial and true values. In this simulation, it is defined as $∥v-v_{0}∥/∥v∥\times \;100$, where $v_{0}$ is the initial value, and *v* is the true value of the velocity vector.

As the initial value deviates further from the true value, the distortion of the velocity vector using the Kalman filter becomes increasingly severe; the improved UKF algorithm does not address these issues perfectly.

## 3. Analysis and Proposed Method

The position and velocity in passive localization have long been addressed as estimation problems. A typical approach involves the use of a KF and improved algorithms. The KF is an optimization method that uses the MMSE criterion. In Figure 2, the KF can be influenced by the initial value; existing filtering methods with the KF framework cannot solve the problems in this study. Thus, we address this issue from the perspective of solving optimization problems.

### 3.1. Gradient Descent Correction

Considering that ${\hat{v}}_{k}$ is the estimated ion of velocity $v_{k}$ at *k* moment, it can be expressed as
(12)$${\hat{v}}_{k}={\left[\begin{array}{ccc} {\hat{v}}_{k}^{x} & {\hat{v}}_{k}^{y} & {\hat{v}}_{k}^{z} \end{array}\right]}^{T}=v_{k}+e_{v}$$
where $v_{k}$ is the true value of the velocity vector, and $e_{v}$ is the estimation error. By introducing ${\hat{v}}_{k}$ into Equation (Equation 5), we can predict observation ${\hat{r}}_{k}$ at the *k* moment:
(13)$${\hat{r}}_{k}={\hat{F}}_{k}{\hat{v}}_{k}$$

The estimation error of the velocity vector can be expressed as
(14)$$E_{k}=r_{k}-{\hat{r}}_{k}=f\left({\hat{v}}_{k}\right)+C$$

In optimizing the velocity vector, the objective function can be expressed as follows, where $R^{3}$ represents the possible value space of $\hat{v}$:
(15)$$\begin{array}{c} \hat{v}=\arg\min_{\hat{v}}∥E∥\\ \;s.t.\;\hat{v}\in R^{3} \end{array}$$

Calculating the gradient of the objective function and correcting the error, we obtain
(16)$$-\nabla E_{k}={\left[-\frac{\partial E_{k}}{\partial v_{x}}-\frac{\partial E_{k}}{\partial v_{y}}-\frac{\partial E_{k}}{\partial v_{z}}\right]}^{T}=-F_{k}^{T}$$

The iterative update of the velocity vector is expressed as follows, where μ is the learning rate:
(17)$${\hat{v}}_{k+1}={\hat{v}}_{k}+\mu E_{k}F_{k}^{T}$$

In this study, the observation for correcting the velocity vector is a one-dimensional scalar; thus, we let μ=1 here and rewrite the gradient descent criterion as a one-dimensional equation for correcting and optimizing the velocity vector mode as shown in Equation (Equation 18):
(18)$${\hat{v}}_{k+1}=F_{k}^{T}\left(∥{\hat{v}}_{k}∥+E_{k}\right)$$

In the model, as Equation (Equation 18) includes the two norms of the velocity vector as the corrected quantity such that $∥F_{k}^{T}∥=1$, we prefer to use the normalized vector as the gradient for the “correction” of the original velocity vector.

### 3.2. Simulated Annealing Mechanism

The gradient descent (GD) [28] used to optimize the objective function often becomes stuck in a local optimal solution. Using Equation (Equation 18) to correct the velocity vector, this problem becomes more severe because there is a lack of observation information. An SA algorithm can solve the problem of finding the local optima using its own mechanism.

Metal reaches a higher energy state if it cools quickly. Based on the analogy of thermodynamics with the cooling and annealing of metals, SA was proposed. When the SA optimizes problems, the objective function to be minimized corresponds to the energy of the states of the metal. SA has become one of many heuristic approaches designed to achieve an optimal solution. This method obtains an optimal solution for a single-objective optimization problem. In the SA, a local minimum is avoided by accepting even worse moves. These moves are not unconditionally accepted, but rather the probabilities are given through a probability function that is normally set as exp(−ΔE/T), where ΔE is the increase in the objective function caused by a worse move; *T* is a control parameter that corresponds to the temperature in the physical annealing analogy. This probability function implies that when ΔE is small or *T* is large, the probability of the solution being accepted is high. As *T* approaches zero, the most worse moves are rejected. Thus, the SA starts at a high temperature to avoid the local minimum and the local optima.

The SA-based algorithm for single-objective optimization is illustrated as follows:Initialize the temperature $T_{0}$.Start with a randomly generated initial solution vector ${\hat{v}}_{0}$ and generate the objective function, such as Equation (Equation 15).Add a random perturbation and generate a new solution vector $\hat{v}\left(k+1\right)$ in the neighborhood of current solution vector ${\hat{v}}_{k}$, and revalue the output ${\hat{E}}_{k}$ of the objective function.If the generated solution vector is archived, make it the current solution vector. Update the existing optimal solution and go to Step 6.Otherwise, accept $\hat{v}\left(k+1\right)$ with the probability exp(−ΔE/T). If the solution is accepted, replace ${\hat{v}}_{k}$ with $\hat{v}\left(k+1\right)$.Decrease the temperature with cooling coefficient α. The new temperature $T=\alpha^{n}T_{0}$, where 0<α<1.Repeat Steps 2–6 until the stopping criterion is met.

The core of the SA optimization process is generating solutions and adding disturbances; the process of generating solutions is not limited. This implies that the current solution or a disturbance can be generated according to different criteria. Thus, the GD can be used in conjunction with the SA mechanism to generate solutions for each move to prevent the GD from falling into a local optimal solution. We add the SA mechanism based on the GD correction to solve the problem of the velocity vector converging to the local optimum (distortion); the results are shown in Figure 3a.

### 3.3. Analysis of Optimization Process

As shown in Figure 3a, after combining the GD correction with the SA mechanism, which is a global optimal search algorithm, the optimization of the velocity vector converges to the true value. However, with a reduction in the error term, the correction step becomes increasingly smaller, requiring a long time for the algorithm to reach the optimal solution. Fortunately, we found that if the optimization or correction criterion for modifying the target is switched on during the process, these issues can be significantly improved. To illustrate this, the models described in this study were simulated. In the simulation, we considered only the error convergence between the values predicted by the model and the actual values. This error was considered only as a numerical value without physical significance. Figure 4 shows the result of combining the GD correction with the SA mechanism to optimize the velocity vector, denoted as GD. For comparison, the results of switching to the EKF during the GD process for further correction of the velocity vector are shown in Figure 4. We chose to switch to EKF earlier during the GD process for comparison, presented in Figure 3b. SA represents the GD optimization with an SA mechanism; EKF represents the solution speed using only the EKF; SA and EKF denote the conditions indicated in Figure 4; CRLB is the Cramer–Rao lower bound.

Comparing Figure 3a and Figure 4, it is evident that switching to the EKF after running a GD with the SA mechanism can converge to a lower error faster and more accurately, which in this model corresponds to the estimation of the velocity vector converging to the global optimal solution. It is believed that switching between different criteria can facilitate optimization of the velocity vector quickly and effectively; the SA mechanism ensures that the optimization process converges to a global optimal solution.

### 3.4. Our Method

From the analysis, we propose an SA optimization algorithm that combines the KF and GD criteria to solve the velocity vector estimation problem in a single-observer passive localization.

As shown in Figure 3, using two distinct optimization criteria can induce changes in the optimization process that are more conducive to finding the global optimal solution (or true value). Thus, we integrated both criteria and assigned them varying weights at different sampling times. Equation (Equation 10) is redefined as
(19)$${\hat{v}}_{k+1}=\left(1-T/T_{0}\right)\left({\hat{v}}_{k}+KE_{k}\right)+T/T_{0}F_{k}\left(∥{\hat{v}}_{k}∥+E_{k}\right)$$

Similarly, the correction of the covariance matrix in Equation (Equation 11) can be rewritten as
(20)$$P=\left(I+\left(1-T/T_{0}\right)KF_{k}+T/T_{0}F_{k}^{T}F_{k}\right)P$$

Unlike the KF, the reformulated Equation (Equation 19) is not an optimal estimate, but rather a suboptimal estimate dynamically weighted by two criteria. Each estimator revision during this suboptimal estimation process deviates from a certain optimal estimation criterion to some extent, akin to random disturbances in the classical SA algorithm. Thus, adding the SA mechanism ensures that the iterative process converges to a global optimal solution.

However, unlike the SA algorithm, our approach is not fundamentally a random estimation but rather a weighted estimation based on different criteria. This renders it impossible for any optimal criterion to constrain each iterative correction, with each correction predicted on multiple criteria. Although it may be impossible to obtain a local optimal correction each time, the correction always aims to reduce errors. This is referred to as a suboptimal estimation (SE).

In this study, the correction of the velocity vector using these two criteria is dynamically weighted. In the SA process, the temperature T decreases gradually; thus, the ratios of temperature to initial temperature $T/T_{0}$ and $1-T/T_{0}$ can be regarded as a group of dynamic normalized weights, expressed in Equation (Equation 20). This also indicates that the strategy proposed at the beginning of the optimization aims for a larger proportion of GD criteria with smaller but more stable corrections, as a relatively mild start is more conducive to the convergence of the algorithm, and an increase in the proportion of revisions based on the MMSE criterion is more conducive to convergence to the true value solution. Our proposed method involves substituting the reformulated equation into the iterative process of the KF, while incorporating the SA mechanism. The resulting pseudocode is integrated into Algorithm 1.
**Algorithm 1** Pseudocode of the proposed algorithm**Require:** Azimuth Angle Vector ($\beta_{1\ldots k}$), Pitch Angle Vector ($\varepsilon_{1\ldots k}$), TOA Observation Value Vector ($r_{1\ldots k}$), Iterations (N), Cooling Coefficient (α), Initial Temperature ($T_{0}$);**Ensure:** Velocity Vector (V);1:$v_{0}$ = RANDOM();2:Initial Q, R, P;3:**for** k = 1 to N **do**4:    $F_{k}$ = [cos$\beta_{k}$cos$\varepsilon_{k}$,sin$\beta_{k}$cos$\varepsilon_{k}$,sin$\varepsilon_{k}$];5:    $T_{k}$ = α$T_{k-1}$;6:    ${\hat{r}}_{k}$ = $F_{k}$$v_{k}$;7:    $E_{k}$ = $r_{k}-{\hat{r}}_{k}$;8:    P = P+Q;9:    K = $PF_{k}^{T}{\left(F_{k}PF_{k}^{T}+R\right)}^{-1}$;10:   $v_{k}$ = $\left(1-T_{k}\right)\left(v_{k-1}+KE_{k}\right)+T_{k}F_{k}\left(|\right|v_{k-1}\left|\right|+E_{k})$;11:   P = $(I+\left(1-T_{k}/T_{0}\right)KF_{k}+T_{k}/T_{0}F_{k}^{T}F_{k})P$;12:   **if** $E_{k}>E_{k-1}$ **then**13:     ρ = RANDOM(0,1);14:     p = $e^{-\left(E_{k}-E_{k-1}\right)/T_{k}}$;15:     **if** ρ>p **then**16:        $v_{k}$ = $v_{k-1}$;17:     **else**18:        $v_{k}$ = $v_{k}$;19:     **end if**20:   **else**21:     $v_{k}$ = $v_{k}$;22:   **end if**23:**end for**24:V = $v_{k}$;

## 4. Simulation

### 4.1. Simulation Scenario

Using the method outlined in Algorithm 1, we simulated velocity vector estimation in a single-observer passive localization. When the relative velocity of two nodes slowly changes, both nodes can be regarded with relatively uniform linear motion within each Δt because Δt is very short. Because Algorithm 1 iteratively approximates the true value based on the observations, the change in velocity does not affect the convergence process. When extreme consideration is given to the presence of high maneuvers between two nodes, the disturbance of acceleration must be considered. The disturbance caused by high maneuvers is represented as process noise in Equation (Equation 8) due to the small value of Δt. Thus, even without prior knowledge (the value of *Q* in Equation (Equation 8)), this small disturbance does not excessively affect the results. In summary, these scenarios for two nodes can be approximated using a uniform linear motion model between the two nodes. When the influence of acceleration cannot be ignored, we assume that the acceleration is constant. The acceleration $\dot{v}$ in this scenario can be decomposed into a constant $\dot{v_{c}}$ and the sum of the perturbations $\ddot{v_{c}}$. The disturbance can be approximated by a uniform linear motion model; thus, only constant acceleration must be considered. Thus, two nodes on the ground and in the air were considered as two scenarios in the simulation: relative uniform motion and relative constant acceleration motion.

### 4.2. Error Distribution

In terms of the noise distribution, this study considered only Gaussian noise. With the assumption that the noise in the KF is Gaussian, satisfactory results cannot be obtained when considering non-Gaussian noise. The optimization strategy is integrated into the KF framework to demonstrate its effectiveness. Thus, the proposed method cannot achieve significantly better results with the KF framework. However, as the most common noise in nature, Gaussian white noise exists in almost all systems, including the observations considered in this study. Any type of colored noise can be represented as the response of Gaussian white noise through a certain system, and can be converted into white noise through whitening filtering. Thus, although this study only considers a Gaussian distribution of noise, it can comprehensively reflect the impact of observation errors generated by noise on the algorithm.

### 4.3. Simulation Parameters

In the following simulation, we considered three scenarios to illustrate the different properties of the proposed method. In each scenario, the main parameters considered are $R_{0}$, the radial vector describing the initial distance between two nodes; *v*, the constant velocity part of the true value of the velocity to be estimated, and *a*, the acceleration part. $\delta v_{0}$ is defined at the end of Section 2, and characterizes the degree to which the initial velocity value deviates from the true value *v*. $\sigma_{r}$, $\sigma_{\beta }$ and $\sigma_{\varepsilon }$ are the standard deviations of the observation error as defined in Equation (Equation 6). *N* is the number of iterations, and Mc is the number of Monte Carlo simulations in the scenario. The specific values of the parameters in the three scenarios are presented in Table 1. The random parameter values are represented by (a,b), such as (0,100); incremental or decreasing values are represented by “to”, such as 0 to 1.

Scenario 1 was used to represent the search process characteristics of the algorithm; thus, many parameters were set to be constant, and the fewest Monte Carlo simulations were required. In Scenario 2, some parameters were randomly selected or changed to demonstrate the effectiveness of our method in a uniform motion model. Scenario 3 added a constant acceleration, which was used to illustrate the effectiveness of our method in a constant acceleration model.

### 4.4. Results and Discussion

In Scenario 1, the velocity vector *v* was set to a fixed value; the EKF and GD were used for comparison with the proposed method. The velocity vector value of each iteration was recorded as a coordinate point in space, with the set of coordinates for each method serving as a group of motion trials. This enabled us to visually observe the search paths of the different methods during the iteration process. As shown in Figure 5, during optimization, both the GD and EKF chose completely different trials. This confirms that the two methods can be viewed as optimizations with different criteria. Although the proposed method can be considered to be weighting, it exhibits a different optimization process. The EKF does not converge to the global optimal solution, which we believe is caused by the low dimensions of observational information. The GD converges to the global optimal solution, but the trial is smooth, indicating that it is a slow process. The optimization process of the proposed method inherits the advantages of both methods and can converge to the globally optimal solution (true value) faster and more directly.

In a more general case, Scenario 2, owing to the random selection of velocity vectors, the error is considered as the relative error, defined as $∥v-v_{k}∥/∥v∥\times 100$.

In practical applications, the number of iterations is related to time. In Equation (Equation 5), the time interval of iteration Δt is normalized, and does not explicitly appear in subsequent equations. The methods considered in the simulation are all based on sampling and updating iterations, sampling the observation data required for each iteration after a time interval, and updating the estimated values. This results in the following relationship between the number of iterations and iteration time:(21)t=NΔt
where *N* is the number of iterations; Δt is the observation interval. For example, when Δt is 0.001 s and the algorithm converges after 100 iterations, the convergence time is 0.1 s. It is observed that both Δt and *N* affect the convergence time of the algorithm. Δt generally depends on the actual observation equipment. This study focuses on the algorithm; thus, Δt is normalized, and only *N* is retained as a measure of the algorithm execution time.

We used the EKF and UKF for comparison. The simulation results are shown in Figure 6a. Examining the velocity error in Figure 6a, it is evident that the UKF, an algorithm better suited for nonlinear filtering than the EKF, can converge to the global optimal solution more accurately than the EKF. However, when compared with the CRLB, the UKF converged solution still exhibits some error with respect to the true value. This issue is not present in the proposed method. Our method quickly converges at a pace similar to that of the EKF and UKF, while also reaching the vicinity of the CRLB.

The root mean square error (RMSE) of the estimate is shown in Figure 6b with increasing $\sigma_{r}$. This error serves as an indicator of the accuracy of the estimator and the iteration stability of the algorithm. In Figure 6b, as $\sigma_{r}$ increases, the RMSE of all methods begins to increase; the UKF always performs better than the EKF, which we believe is attributable to the resampling of the unscented transform. The proposed methods have significantly higher estimation accuracies as shown in Figure 6a; they converge more closely to the CRLB.

In Figure 6a, in the early iteration process before approximately 80 iterations, the error of the UKF is the closest to that of the CRLB. However, in an actual scenario, the extent to which the current iteration speed differs from the true value is unknown. Thus, when estimating, the convergence of the method must first be confirmed, which implies that the results remain almost unchanged after a certain number of iterations. When the data are constantly corrected, even if the value of a certain iteration is closer to the true value, it cannot be detected because the true value is unknown. Thus, the stable value of the algorithm is generally considered the convergence value of the algorithm. Thus, although the UKF is closer to the CRLB in the early stages of the iteration process, this feature cannot be effectively utilized.

To further elaborate the benefits of our proposed method, we present the cumulative probability distribution (CDF) of these methods when $\sigma_{r}$ = 0.5 as shown in Figure 7. The proposed method has smaller error values and tends to have a smaller error distribution.

In Scenario 3, when the acceleration cannot be ignored, the observation model given in Equation (Equation 4) begins to fail because the influence of acceleration is not considered in the model. This has a fatal impact on the velocity vector estimation, equivalent to the presence of an imperceptible influencing factor. However, for the proposed method, the SE optimization strategy can result in an estimation closer to an instantaneous velocity vector. Each iteration is corrected for changes caused by acceleration.

Figure 8a,b show the RMSE and CDF, respectively, when the acceleration remained constant at 2.5 m/s${}^{2}$. The model errors were fatal to both the EKF and UKF; their results are extremely poor. In contrast, with the SE optimization strategy, our method can accurately track the real-time speed and converge to a stable error level similar to that in a uniform motion model. This indicates that our method has a higher estimation accuracy and is extremely robust, owing to the heuristic nature of the SE optimization strategy.

## 5. Conclusions

This study examined the independent estimation of the velocity vector in a passive localization scenario and proposed a suboptimal estimation method. Our proposed method considers the estimation of the velocity vector as a nonlinear filtering problem and as an optimization problem. The optimization process uses dynamic weighting of the linear optimal estimation and GD correction, with an SA mechanism used to reconcile the two criteria and ensure convergence. The experimental results for the two models demonstrate that when the observed dimension is lower than the estimated dimension and there is insufficient prior knowledge, the proposed suboptimal estimation strategy converges quickly and accurately to the target value of the velocity vector. In addition, when there is a disturbance from acceleration, the proposed suboptimal estimation strategy demonstrates strong robustness. We intend to explore the combination of a suboptimal estimation strategy and neural networks to further improve the performance of single-observer passive localization.

## Figures and Tables

**Figure 1 sensors-23-05940-f001:**
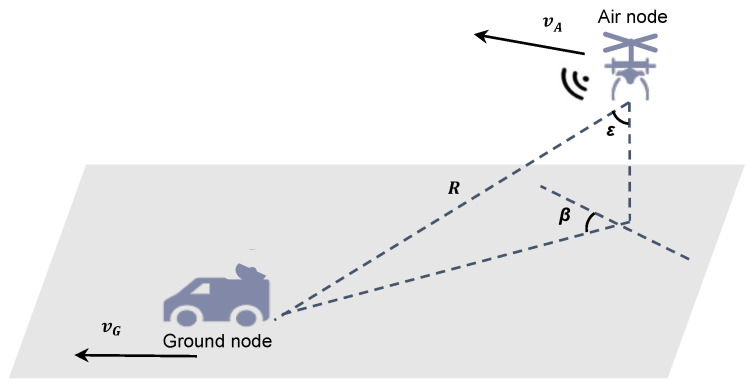
Schematic diagram of the single-observer passive localization discussed in this paper.

**Figure 2 sensors-23-05940-f002:**
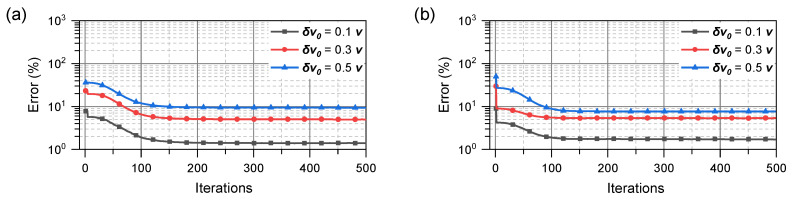
Performance of (**a**) EKF; (**b**) UKF with different initial values.

**Figure 3 sensors-23-05940-f003:**
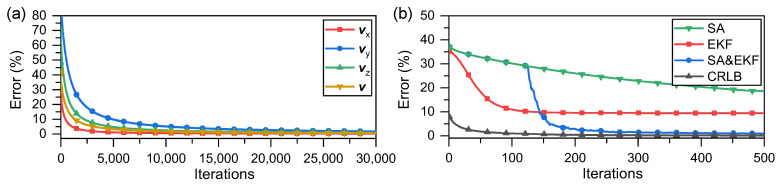
(**a**) Velocity optimization results after adding SA mechanism; (**b**) comparison of results of different methods with CRLB.

**Figure 4 sensors-23-05940-f004:**
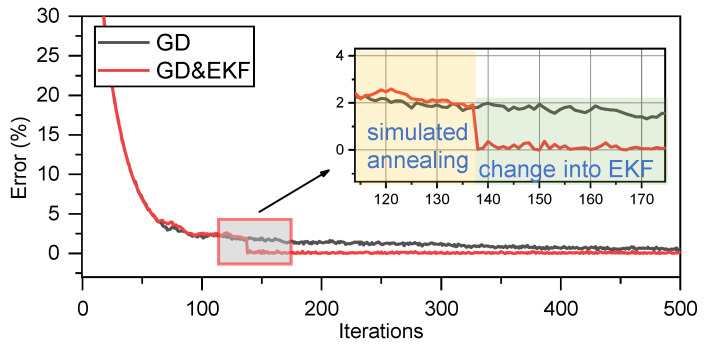
Switching to EKF when GD corrects velocity vector.

**Figure 5 sensors-23-05940-f005:**
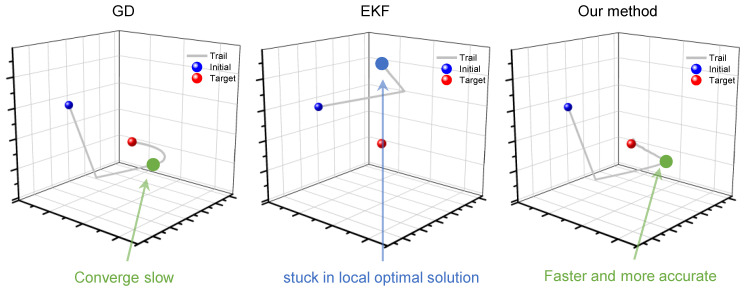
Comparison of speed vector search trials for three methods in Scenario 2.

**Figure 6 sensors-23-05940-f006:**
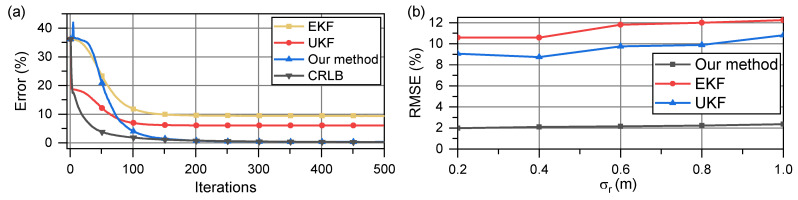
(**a**) Comparison of velocity vector estimation error and CRLB when two nodes move at a relatively uniform velocity; (**b**) RMSE of all methods when $\sigma_{r}$ ranges from 0.2 m to 1 m.

**Figure 7 sensors-23-05940-f007:**
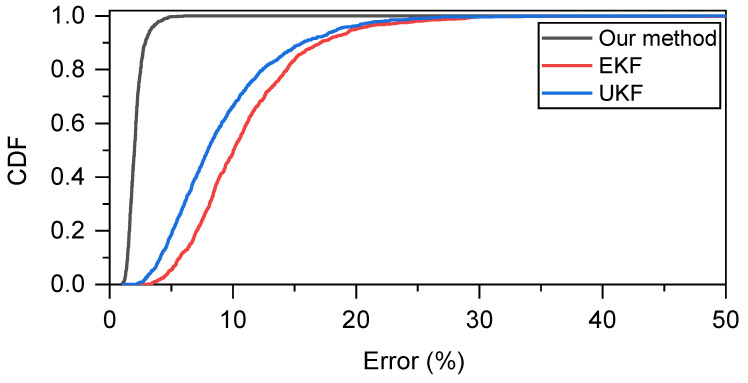
Comparison of CDF when $\sigma_{r}$ = 0.5 when two nodes move at relatively uniform velocity.

**Figure 8 sensors-23-05940-f008:**
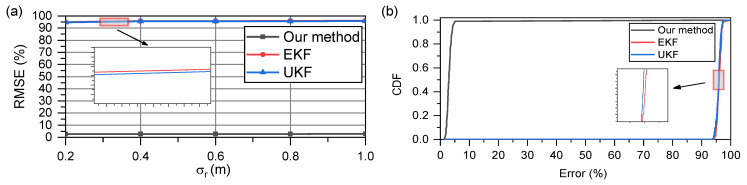
With two nodes moving relatively at an acceleration of 2.5 m/s${}^{2}$: (**a**) RMSE of all methods when $\sigma_{r}$ ranges from 0.2 m to 1 m; (**b**) comparison of CDF when $\sigma_{r}$ is 0.5.

**Table 1 sensors-23-05940-t001:** Simulation parameters.

Scenario	$R_{0}$	*v*	$\delta v_{0}$	*a*	$\sigma_{r}$	$\sigma_{\beta }$	$\sigma_{\varepsilon }$	*N*	Mc
Scenario 1	[0.5, 0.5, 0.2] km	[92, 86, 5] m	0.1	0 m/s${}^{2}$	0.2 m	$5^{\circ }$	$5^{\circ }$	1000	100
Scenario 2	[(0, 1), (0, 1), (0, 1)] km	[(0, 0.1), (0, 0.1), (0, 0.1)] km/s	(0.3, 0.5)	0 m/s${}^{2}$	0.2 to 1 m	$5^{\circ }$	$5^{\circ }$	500	1000
Scenario 3	[(0, 1), (0, 1), (0, 1)] km	[(0, 0.1), (0, 0.1), (0, 0.1)] km/s	(0.3, 0.5)	2.5 m/s${}^{2}$	0.2 to 1 m	$5^{\circ }$	$5^{\circ }$	500	1000

## Data Availability

No publicly available data.

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
