# Peer review of "A Suboptimal Optimizing Strategy for Velocity Vector Estimation in Single-Observer Passive Localization"

_sensors, 2023, doi:10.3390/s23135940_

Round 1

Reviewer 1 Report

While the article is well structured, language revision is highly recommended. There are instances with text being repeated text and multiple grammatical/stylistic errors.

Author Response

Thank you so much for your kind comments on our research work. We have addressed the following constructive comments point by point and carefully revised the original manuscript accordingly (modifications are marked in blue). The point-to-point response to the referee has also attached named "response_to_referee.pdf". 

Reviewer 2 Report

Some minor suggestions are as follows:

1-    It is better to include a line under the equation saying what the variables mean. For instance, it is better to read equation (5) as

When one node observes the radiation signal of another node at constant intervals ??, the radial distance at moment ? − 1 between two nodes can be noted as follow:

                               Equation (5),

where v_k denotes the velocity vector between two nodes at time k.

2-     Is the assumption on observation noise correct in equation (8)? How about non-Gaussian noise? If so, how is accuracy of your method?

Author Response

We sincerely thank for the reviewer’s patient reading and kind comments on our manuscript.  We hope our revisions can make readers clearly understand the working mechanism and application of our devices based on ferroelectric superdomain controlled graphene plasmons. For your guidance, itemized responses to comments and other changes in the context are appended below. All changes made to the text are in blue. The point-to-point response has also attached which is named "response_to_referee_2.pdf".

Round 2

Reviewer 1 Report

The authors have done a good job addressing comments made to the original version of the manuscript. The updated article has a good structure, and much better flow with the new details added. However, there are still several details that should be improved:

There appear to be numerous typos/errors in the highlighted sections of the text, particularly on pages 3 and 4 of the document. It must be reviewed and edited. 

Page 3, line 64… lLocalization

Page 3, line 64 ...However,But…

Page 3, line 65…node;.

Page 3, line 67… change inof the radial vector…

Page 3, line 68… atthe change inof the radial vector

Etc. etc.

Same comment to the non-highlighted text in the updated version of the manuscript. Numerous typos present, potentially introduced by the language correction tool used by the authors.

I find the authors' comment that experimental evaluation is very challenging to carry out completely reasonable. It was not my expectation that field test results would be available. I think additional simulation results broadened the presented analysis and allowed for a more critical discussion of the approach. 

The updated manuscript requires a thorough language review. There are numerous typos/errors in both the old and highlighted sections of the text potentially introduced by the language correction tool used by the authors.

Author Response

Response to the comments of Reviewer #1:

General comments: The authors have done a good job addressing comments made to the original version of the manuscript. The updated article has a good structure, and much better flow with the new details added. However, there are still several details that should be improved:

There appear to be numerous typos/errors in the highlighted sections of the text, particularly on pages 3 and 4 of the document. It must be reviewed and edited.

Page 3, line 64… lLocalization

Page 3, line 64 ...However,But…

Page 3, line 65…node;.

Page 3, line 67… change inof the radial vector…

Page 3, line 68… atthe change inof the radial vector

Etc. etc.

Same comment to the non-highlighted text in the updated version of the manuscript. Numerous typos present, potentially introduced by the language correction tool used by the authors.

I find the authors' comment that experimental evaluation is very challenging to carry out completely reasonable. It was not my expectation that field test results would be available. I think additional simulation results broadened the presented analysis and allowed for a more critical discussion of the approach.

Response: Dear reviewer, we sincerely thank your affirmation of our revised manuscript and your careful reading. We feel sorry that the numerous typos due to our negligence. We have rechecked the manuscript based on your suggestion and corrected any spelling and punctuation errors that have occurred. We marked the revised paragraphs in blue in the latest manuscript.

More important, for your kind comments on the additional simulations, we are actively looking for co-workers and will systematically investigate this for your useful suggestion. We will report the latest discoveries in time once. Thanks again for your useful comments and we look forward to your guidance if possible.